# Effect of Storage Conditions on the Quality of Arbequina Extra Virgin Olive Oil and the Impact on the Composition of Flavor-Related Compounds (Phenols and Volatiles)

**DOI:** 10.3390/foods10092161

**Published:** 2021-09-13

**Authors:** Leeanny Caipo, Ana Sandoval, Betsabet Sepúlveda, Edwar Fuentes, Rodrigo Valenzuela, Adam H. Metherel, Nalda Romero

**Affiliations:** 1Departamento de Ciencia de los Alimentos y Tecnología Química, Facultad de Ciencias Químicas y Farmacéuticas, Universidad de Chile, Santiago 8380000, Chile; leeannyc@gmail.com (L.C.); anamsv.as@gmail.com (A.S.); 2Centro Para el Desarrollo de la Química, CEPEDEQ, Facultad de Ciencias Químicas y Farmacéuticas, Universidad de Chile, Santiago 8380000, Chile; bsepulveda@ciq.uchile.cl; 3Departamento de Química Inorgánica y Analítica, Facultad de Ciencias Químicas y Farmacéuticas, Universidad de Chile, Santiago 8380000, Chile; edfuentes@ciq.uchile.cl; 4Departamento de Nutrición, Facultad de Medicina, Universidad de Chile, Santiago 8380000, Chile; rvalenzuelab@med.uchile.cl; 5Department of Nutritional Sciences, University of Toronto, Toronto, ON M5S1A8, Canada; adam.metherel@utoronto.ca

**Keywords:** olive oil, quality, storage conditions, phenols, volatile compounds, (*E*)-2-nonenal

## Abstract

Commercialization of extra virgin olive oil (EVOO) requires a best before date recommended at up to 24 months after bottling, stored under specific conditions. Thus, it is expected that the product retains its chemical properties and preserves its ‘extra virgin’ category. However, inadequate storage conditions could alter the properties of EVOO. In this study, Arbequina EVOO was exposed to five storage conditions for up to one year to study the effects on the quality of the oil and the compounds responsible for flavor. Every 15 or 30 days, samples from each storage condition were analyzed, determining physicochemical parameters, the profiles of phenols, volatile compounds, α-tocopherol, and antioxidant capacity. Principal component analysis was utilized to better elucidate the relationships between the composition of EVOOs and the storage conditions. EVOOs stored at −23 and 23 °C in darkness and 23 °C with light, differed from the oils stored at 30 and 40 °C in darkness. The former was associated with a higher quantity of non-oxidized phenolic compounds and the latter with higher elenolic acid, oxidized oleuropein, and ligstroside derivatives, which also increased with storage time. (*E*)-2-nonenal (detected at trace levels in fresh oil) was selected as a marker of the degradation of Arbequina EVOO quality over time, with significant linear regressions identified for the storage conditions at 30 and 40 °C. Therefore, early oxidation in EVOO could be monitored by measuring (*E*)-2-nonenal levels.

## 1. Introduction

Olive oil is highly valued by consumers for its nutritional and health properties, as well as for its sensory attributes [1]. According to the Trade Standard Applying to Olive Oils and Olive Pomace Oils [2], virgin olive oil is classified as extra virgin olive oil (EVOO) if it meets certain criteria for free acidity, peroxide value, and absorbance of ultra-violet light, among other parameters. Likewise, it must also comply with certain organoleptic characteristics, the median of defects equal to zero and the median of the fruity attributes greater than zero.

The Australian Standard [3] and the Best Practice Guidelines for the Storage of Olive Oils [4] recommend even in the case of top quality oils at production, kept in the strictest storage conditions, limit the best before date to 24 months after bottling, during which it is expected the product retains its specific properties and preserve its ’extra virgin’ category. However, depending on the storage conditions, the quality of EVOO may vary. An oil stored in inadequate temperature, light, or oxygen conditions can be affected in its quality parameters [5]. Moreover, its minor components, such as phenolic compounds responsible for the bitter and pungent taste of olive oil [6] or volatile compounds responsible for the aroma [7], can be negatively affected by a series of chemical reactions that favor oxidative processes in the oil [8]. The formation of undesirable compounds such as ketones, aldehydes, volatile acids [8], and oxidized phenols [9] can alter the characteristic flavor of EVOO [8] and be detrimental to its nutraceutical properties [1].

Several studies have demonstrated the effects of storage conditions on oil phenols, reporting a decrease in total phenolic content, oxidation of secoiridoid derivatives and increases in hydroxytyrosol and tyrosol, among others [1,9,10,11,12,13]. Although a specific phenolic content or composition is not included in the IOC Trade Standard [2], a decrease in its content or significant changes in its composition can affect the stability of olive oils [6,14]. These changes favor the oxidation of fatty acids, and with it the formation of undesirable compounds, which results in the appearance of off-flavors [8]. As a result, the oil may lose its extra virgin category with a consequent reduction in product value and consumer acceptability [1].

An important aspect of EVOO quality to consider is what happens to the oil once it is marketed. Although guidance has been given to manufacturers [4] on the best storage conditions for oil to minimize its degradation, they cannot control what happens on the supermarket shelf. This is even more relevant for products exported to markets with higher average temperatures than those of the country of origin. Consequently, discrepancies may exist between the actual quality of the product in the supermarket, the quality declared on the label, and the expected by the consumers in the purchase it [1].

While several studies have been published on the effects of storage conditions (light, temperature, oxygen, time, type of packaging) on the quality of EVOO or VOO [5,15,16,17,18], few have also focused on the effects of storage conditions on the composition of phenolic or volatile compounds. Moreover, information on the effects of storage conditions on both types of compounds is scarce, and most of the available studies have been conducted at room temperature between 6 and 25 °C in darkness [9,10,11,12,13,19,20,21,22,23,24,25,26]. Thus, the aim of this study was to investigate the effects of five storage conditions on the quality parameters of Arbequina variety EVOO and how these storage conditions impact flavor-related compounds, particularly phenols and volatiles.

## 2. Materials and Methods

### 2.1. Reagents

All reagents, either analytical or HPLC grade, were purchased from Merck (Darmstadt, Germany). The phenol standards (3-hydroxytyrosol, 2-(4-hydroxyphenyl) ethanol (tyrosol), p-coumaric acid, vanillic acid, vanillin, luteolin, apigenin, pinoresinol, p-hydroxyphenylacetic acid (internal standard 1), o-coumaric acid (internal standard 2) oleuropein and caffeic acid), Trolox, fluorescein, and 2,2′-Azobis(2-amidinopropane) dihydrochloride (AAPH) were obtained from Sigma-Aldrich (St. Louis, MO, USA). Volatile standards 4-methyl-2-pentanol (internal standard), ethanol, ethyl propionate, 4-methyl-2 pentanone, butyl acetate, 2-methyl-1-butanol, 3-methyl-1-butanol, 3-octanone, acetic acid, propionic acid, 1-octanol, butyric acid, heptanoic acid, nonanoic acid, (*E*)-2-hexenal, hexanal, hexanol, (*E*)-2-nonenal were purchased from Merck. Tocopherols standards were purchased from Calbiochem (Merck). All standards had a purity of 98% or higher.

### 2.2. Sample Preparation and Sampling

The study was performed with Arbequina variety EVOO from Huasco valley, harvested in 2017, which was purchased from Payantume. The Arbequina oil presented extra virgin quality according to official analytical methods and limits (free acidity < 0.8% in oleic acid, K232 < 2.50, K270 < 0.22, ∆K < 0.01; [2]. In addition, the oil contained 17.3% saturated fatty acids, 70.9% monounsaturated fatty acids, and 11.8% polyunsaturated fatty acids.

#### 2.2.1. Storage Conditions

Given that EVOO has expanded to markets with higher average temperatures than those of the countries of origin and that the studies published have mostly been carried out at temperatures between 6 and 25 °C in darkness, this work included higher storage temperatures with the purpose of visualizing the problems of these new markets. Another factor to study was the effect of light versus darkness at room temperature since the oxidation mechanisms are different, especially in oils that contain chlorophyll pigments. Thus, 5 storage conditions were studied: Condition 1 or control condition (C1): −23 ± 2 °C in darkness; Condition 2 (C2): room temperature (23 ± 2 °C) with light; Condition 3 (C3): room temperature (23 ± 2 °C) in darkness; Condition 4 (C4): 30 ± 2 °C in darkness; Condition 5 (C5): 40 ± 2 °C in darkness.

#### 2.2.2. Preparation of Samples Stored in Darkness

The oils were aliquoted into amber glass jars of 100 mL capacity, with 3% (*v*/*v*) headspace. The headspace of the jars was purged with nitrogen to eliminate oxygen during storage, and the jars were closed with an airtight lid. The samples were stored in triplicate at each of the four storage temperatures either in a freezer (C1), at room temperature in closed shelves (C3) or in an oven (C4 and C5).

#### 2.2.3. Preparation of Samples Stored at Light

The samples stored with light (C2) were aliquoted in triplicate into clear glass jars of 100 mL capacity under the same conditions as samples stored in amber glass jars. These were then placed on shelves and exposed to artificial light intensity of ≈1870 lx, exposing them to 12 h of light and 12 h of darkness.

#### 2.2.4. Sampling

Conditions C1 and C5 were sampled every 3 months and 15 days, respectively. Conditions C2, C3, and C4 were sampled monthly. The study was carried out for 1 year, and samples were analyzed monthly in triplicate using the analytical methods described below.

### 2.3. Quality Parameters

Free fatty acids (Ca 5a-40), peroxide value (Cd 8-53), and specific extinctions of oils (K232, K270, ΔK) (Ch 5-91) were determined according to AOCS standard methods [27]. 

### 2.4. Determination of Phenolic Compounds

The phenol composition was determined according to Fuentes et al. [28]. p-hydroxyphenylacetic acid was used as an internal standard for the quantification of phenolic compounds (other than flavones and ferulic acid) and elenolic acid at 280 and 235 nm, respectively. In addition, o-coumaric acid was used as an internal standard to the quantification of flavones (luteolin and apigenin) and ferulic acid at 335 nm. The results were expressed in mg/kg.

### 2.5. Total Phenolic Content

Phenolic compounds were extracted according to IOOC [29] with modifications according to Fuentes et al. [28]. 

### 2.6. Hydrophilic Orac Assay (H-ORACFL)

ORACFL assays were performed according to those described by Fuentes et al. [28] using 0.075 M phosphate buffer (pH 7.4). The antioxidant capacity was expressed as μmol Trolox equivalent (TE)/g oil.

### 2.7. Tocopherol Content

The standard method AOCS Ce 8-89 [27] was used to determine tocopherols levels by high-performance liquid chromatography (HPLC). Details of the methodology were published in Fuentes et al. [27]. The chromatographic signals were processed by Clarity chromatographic software (DataApex, Prague, The Czech Republic). Alpha-tocopherols were identified and quantified using Calbiochem α-tocopherol (Merck, Darmstadt, Germany) as an external standard. The results were expressed in mg/kg.

### 2.8. Volatile Compounds

The EVOO samples (2 g), spiked with 100 mg of internal standard solution of 4-methyl-2-pentanol dissolved in refined sunflower oil (50 µg/mL), were placed in a 20 mL glass vial, tightly capped with a polytetrafluoroethylene (PTFE) septum, and held for 5 min at 40 °C to allow for the equilibration of the volatiles in the headspace. After the equilibration time, the septum covering each vial was pierced with a solid-phase microextraction (SPME) needle, and the fiber was exposed to the headspace for 40 min. When the process was completed, the fiber was inserted into the injector port of the GC. The temperature and time of the pre-concentration step, performed in a HT280T (HTA s.r.I, Brescia, Italy), were automatically controlled by the software HT-COMSOFT (HTA s.r.I). The SPME fiber (2 cm length and 50/30 μm film thickness) was from Supelco (Bellefonte, PA, USA) and consisted of a stable flex stationary phase divinylbenzene/carboxen/polydimethylsiloxane (DVB/CAR/PDMS). The fiber was previously conditioned following the instructions of the supplier.

#### Determination of Volatile Compounds

The volatiles absorbed by the fiber were thermally desorbed in the hot injection port of a GC for 5 min at 260 °C with the purge valve off (splitless mode) and were then injected onto a TR-WAX capillary column (60 m × 0.25 mm i.d., 0.25 μm coating; Teknokroma, Barcelona, Spain) of a Shimadzu GC-2010 Plus gas chromatograph with a flame ionization detector (FID) (Shimadzu, Kyoto, Japan). The carrier gas was hydrogen with a flow rate of 1.5 mL/min. Detector temperature was 280 °C, and the oven temperature was held at 40 °C for 10 min and then programmed to increase by 3 °C/min to a final temperature of 200 °C, where it was held for 10 min. The data were recorded and processed with GC solution Ver. 2 Workstation Software (Shimadzu, Kyoto, Japan). Each sample was analyzed in triplicate. The identification of the volatile compounds was performed by comparison of the retention times with Sigma standards and quantified by interpolation on the calibration curves. The calibration curves were constructed using sunflower oil spiked with volatile compounds standard (Sigma) in a range from 10 to 1000 µg/mL. The linear regression coefficients (R) were in a range from 0.975 to 0.994.

### 2.9. Statistical Analysis

The results were presented as means ± standard deviation. The data were statistically analyzed using an unpaired Student’s t-test and one-way ANOVA to compare the means and a Mann–Whitney test to compare the medians. In all statistical tests *p*-values lower than 0.05 were considered statistically significant. The analyses were performed using the Statgraphic XV software (Rockville, MD, USA). The multivariate general characterization of the samples, considering all the physicochemical parameters determined, was performed by principal component analysis (PCA) using the Unscrambler software (CAMO PROCESS AS, Oslo, Norway). Partial least squares (PLS) were selected to build a model to identify how the Y variable behaves as a function of the independent variables, rather than used for future predictions. The X matrix corresponds to the chemical characterization, the Y column corresponds to the (*E*)-2-nonenal, and the B matrix corresponds to the regression vector. This vector characterizes the influence of each variable in the model. The same software was used for PLS analysis. The optimum number of factors to be used within the PLS regression and PCA was determined through a full cross-validation procedure, which consists of systematically removing one of the training samples, in turn, and using only the remaining ones for construction of the latent factors and/or regression coefficients. All data were previously centered.

## 3. Results and Discussion

### 3.1. Quality Parameters

Quality parameters, including free fatty acids, PV, K232, and K270 measured in Arbequina variety EVOO subjected to five storage conditions of light and temperature for 12 months are reported in Table 1. A slight increase in free fatty acids (%FFA) over time was observed in C1 (−23 ± 2 °C in darkness), C2 (23 ± 2 °C with light), and C3 (23 ± 2 °C in darkness) storage conditions, however, values between 0.17–0.20% of oleic acid were relatively stable. The %FFA of samples stored at C4 (30 ± 2 °C) and C5 (40 ± 2 °C) conditions, both in darkness, increased beginning after 6 months, reaching a statistically higher value (*p* < 0.05) of 0.32 % of oleic acid in the C5 condition compared to all other storage conditions. The increase in %FFA at higher temperatures might be due to higher hydrolytic activity of lipase enzymes on triacylglycerols that remain in the oil after extraction [5], which could be further influenced by the moisture levels in the oil [30]. In all conditions, the %FFA did not reach 0.8%, the maximum allowed by IOC for EVOO category [2], and were lower than those reported by Ayton et al. [5] when olive oil was stored at 37 °C without oxygen in darkness.

Peroxide value (PV) is an indicator of initial oxidation and is considered to be a relevant quality index because it detects the oxidation before it is perceived organoleptically [31]. As shown in Table 1, the fresh EVOO yielded a low peroxide value (2.9 meq O_2_/kg), indicating that the oil was processed in good manufacturing conditions. The C1 condition (control) showed little to no change during storage time, reaching a value of 3.3 meq O_2_/kg of oil. The highest PV was reached in the C2 condition after 12 months of storage (8.2 meq O_2_/kg of oil), compared with the C3, C4, and C5 conditions with values of 5.9, 5.3, and 4.4 meq O_2_/kg of oil, respectively. The largest increases in PV were observed during the first 3–4 months of storage when hydroperoxides were formed with the remaining oxygen in the oil. In the C5 condition, a decrease in PV from the third month of storage was observed, which could be explained by the decomposition of hydroperoxides and subsequent formation of products of secondary oxidation. Similar results were published by Esposto et al. [26], who attributed a low PV accumulation to the rapid conversion of hydroperoxides to related products, such as C7–C11 volatiles, responsible for rancid defect in oil. Furthermore, Ayton et al. [5] observed a decrease in the PV of olive oil stored without oxygen in darkness, particularly when stored at 37 °C.

The higher level of the hydroperoxides formed in the C2 condition during storage could be explained by the action of light on the oils chlorophyll, or photooxidation. Light-activated chlorophyll would act on oxygen at fundamental state to generate singlet oxygen, which would then react with unsaturated fatty acids to form hydroperoxide [31]. The darkness conditions C3, C4, and C5, would follow the classic autoxidation mechanism with the formation of free radicals from an unsaturated fatty acid, such as linoleic acid. In this study, light appeared to have more impact than temperature on PV, suggesting that photooxidation is more effective than autoxidation in the formation of hydroperoxide in olive oil.

The oxidation of unsaturated fatty acids resulting from the formation of peroxyl radicals and hydroperoxides alters the configuration of the double bonds, transforming the normal configuration of interrupted methylene into its conjugated forms. These conjugated compounds absorb light at 232 nm (K232) and are indicative of primary oil oxidation [32]. Table 1 shows a marked increase in K232 up to 3 or 4 months of storage, with C4 and C5 conditions yielding the largest increases during this period. After 4 months, K232 values remain relatively stable until 12 months of storage, reaching values of 2.02 and 1.96 for the C4 and C5 storage conditions, respectively. The storage condition with light exposure, C2, showed only a moderate increase in K232, most likely due to the rapid decomposition of hydroperoxides where the breakdown of the conjugated dienes would give way to the formation of secondary oxidation products. Condition C3 demonstrated fluctuating K232 values, reaching levels similar to C4 and C5, while condition C1 (control) remained stable. This fluctuation of the K232 values could be explained by the decomposition of hydroperoxides, as noted previously.

K270 measures secondary oxidation products (aldehydes and ketones) produced by the breakdown of hydroperoxides, reflecting a more advanced oxidation state than PV or K232 [33]. Fresh oil had a low K270 value of 0.09 and C1 (control) remained unchanged. However, when evaluating the effect of light on the formation of secondary oxidation products, storage at room temperature with light (C2) resulted in a higher formation of these compounds, reaching a K270 value of 0.15. Conversely, storage at room temperature in darkness (C3) remained relatively stable. When evaluating the effect of temperature (in darkness), it was observed that storage at 40 °C (C5) yielded the highest formation of secondary oxidation products, and surpassed all the other conditions by reaching a K270 value of 0.18. In support, the study by Ayton et al. [5] noted that the effect of temperature was significant in the formation of secondary oxidation products.

### 3.2. Behavior of Alpha Tocopherol during Storage

Alpha-tocopherol is the main fat-soluble antioxidant present in olive oil. The initial α-tocopherol content in the Arbequina variety EVOO was 180 mg/kg (Table 2), similar to those published by Fuentes et al. [28]. Figure 1 shows the degradation of α-tocopherol for all conditions during 12 months of storage and highlights the dramatic drop of α-tocopherol in the C2 condition throughout the storage period, reaching 80 mg/kg after 12 months, equivalent to a retention percentage of 40%. These results were similar to those reported by Ayton et al. [5] in olive oil stored at 22 °C with light and without oxygen in which they reported a rapid decline of α-tocopherol during the first 6 months of storage, losing a total of 50% of the initial content after 36 months of storage.

Conditions C1, C3, C4, and C5 presented smaller losses of α-tocopherol with retention percentages of 80, 71, 69, and 65%, respectively. The greatest degradation of α-tocopherol at room temperature with light (C2) is likely attributed to the effect of light on the chlorophylls and its degradation products of EVOO. As previously discussed, these degradation products would act as photosensitizers, activating oxygen at a fundamental state to form singlet oxygen, accelerating the oxidation of the oil by photooxidation. Alpha-tocopherol would act by trapping singlet oxygen via a charge transfer attenuation mechanism [34], resulting in the production 8-hydroperoxy-tocopherone, which is easily split by mild acidic conditions to tocopherylquinone, causing the loss of α-tocopherol [35]. 

Temperature also appears to strongly influence the degradation of α-tocopherol as storage at 40 °C (C5) resulted in significant degradation (*p* < 0.05) relative to the initial level, despite being stored in darkness. Under this condition, α-tocopherol would interrupt the propagation step in lipid autoxidation based on the redox system of tocopherol-tocopheryl semiquinone [34,35]. The large loss of α-tocopherol in C2 can be attributed to the greater efficiency of photooxidation on lipid oxidation. 

### 3.3. Behavior of Phenolic Compounds during Storage

Table 2 shows the changes in levels of phenolic compounds following 12 months of storage. Arbequina variety EVOO had a total phenol content of 247 mg/kg of oil at the beginning of the storage period. This total phenol amount is in the expected range found for this variety and depends on other variables such as agroclimatic and processing factors [28]. 

As shown in Table 2, the simple phenols tyrosol and hydroxytyrosol, and elenolic acid increased during storage as a likely result of the hydrolytic degradation of the secoiridoid compounds [36]. The largest increases were observed after 12 months of storage at room temperature with light (C2), and at higher temperatures in darkness (C4 and C5). The increase in tyrosol and hydroxytyrosol was not proportional to that of elenolic acid (molar ratio), possibly due to the active role of both phenols as antioxidants, which can trap free radicals formed under different storage conditions. Castillo-Luna et al. [9] proposed hydroxytyrosol as a reliable marker to detect aged EVOOs and blends prepared with aged EVOOs.

Following 12 months of storage, no significant change was observed in the content of total phenols (*p* > 0.05). Nevertheless, a significant increase in the content of total oxidized phenols was detected under conditions C2, C3, C4, and C5, reaching a value of 63 mg/kg in the latter, equivalent to 24% of the total phenols. The increase in oxidized phenols may be due to the oxidation of secoiridoid compounds, oleuropein, and ligustroside derivatives, predominant compounds present in fresh EVOO [37], which were oxidized at between 10 and 50% relative to their initial content. 

The decarboxymethylated compounds derived from oleuropein (oleacein) and ligustroside (oleocanthal) were the most oxidized (Table 2). This large degradation could be explained by the chemical structure and reactivity of secoiridoid derivatives. Oleacein and oleocanthal, with two aldehydic functional groups in an open configuration, possess larger reactivity than monoaldehyde oleuropein aglycone isomers, which are mainly present in a closed configuration [9]. Tsolakou et al. [12] has proposed oleocanthalic acid formed by oxidation of oleocanthal as a marker of EVOO aging.

In a study of EVOO storage in darkness and room temperature for 24 months, Kotsiou and Tasioula [11] reported that the decrease in secoiridoid derivatives gives rise to simple phenols (hydroxytyrosol and tyrosol) and the formation of oxidized products and attributed these changes to hydrolytic and oxidative effects, respectively.

Figure 2 shows the degradation of the total secoiridoid compounds, of the decarboxymethylated forms, oleacein, and oleocanthal, and the formation of oxidized phenols, under C2 (Figure 2a), and C5 (Figure 2b) conditions. The C5 condition resulted in the highest degradation of these compounds, with temperature seemingly being the preponderant factor. Levels reached 48 and 35% of retention percentage for total secoiridoid compounds and decarboxymethylate derivatives, respectively. Secoiridoid compounds and the decarboxymethylated forms, oleacein and oleocanthal, resulting from the C5 condition followed a kinetic degradation of zero-order (k = −6.3491 ± 0.7157 mg/kg × months^−1^; r^2^ = 0.9608), (k = −5.8785 ± 0.5923 mg/kg × months^−1^; r^2^ = 0.9629), respectively, with no significant differences (*p* > 0.05) between them. In the C2 condition, the largest degradation was observed in decarboxymethylated derivatives, reaching a retention percentage of 59% at 12 months of storage, while secoiridoid compounds reached retention percentage values of 71%. The highest levels of oxidized phenols were present in the highest temperature condition (C5), with a value of 63 mg/kg. Castillo-Luna [9] reported degradation of total phenols in a range of 22 to 61% in EVOO subjected to 20 °C in darkness for 12 months. Another study conducted by Montesano et al. [38] reported a 40% decrease in tyrosol equivalents content in EVOO following 28 weeks of storage at 20 °C in darkness.

Phenolic compounds together with α-tocopherol would function as a strong antioxidant system in EVOO, trapping the free radicals of unsaturated fatty acids formed by both autoxidation and photooxidation. Alpha-tocopherol appears to have a preponderant role in photooxidation (C2 condition), as demonstrated by its higher levels of degradation when exposed to light. Conversely, phenolic compounds appear to have a more active role in autoxidation, given by the larger degradation and the lesser degradation of α-tocopherol in the C5 condition. A possible interaction between α-tocopherol and phenolic compounds cannot be ruled out. 

### 3.4. Antioxidant Capacity

Table 2 shows the antioxidant capacity of Arbequina EVOO in fresh oil and after 12 months of storage. The fresh oil presented an ORAC value of 5.5 μmol TE/g of oil, lower than that presented by Fuentes et al. [28] in the same variety. A significant decrease (*p* < 0.05) in antioxidant capacity was observed in all storage conditions, ranging from 3.6 to 4.1 µmol TE/g of oil after 12 months of storage, with no significant differences (*p* > 0.05) between conditions.

Sanmartin et al. [39] reported decreases of 10 and 15% in antioxidant capacity in EVOO samples stored at 6 and 26 °C, respectively, in tinplate and greenish glass containers for 125 days. Since the antioxidant capacity depends on the concentration of the phenol and its structure [40], the decrease in the antioxidant capacities of the oils may be related to the decrease in non-oxidized phenols and the change in the profile of the phenolic compounds with storage time.

### 3.5. Behavior of Volatile Compounds during Storage

Table 3 presents the composition of volatile compounds of EVOO at baseline and after 12 months of storage for the five conditions. The profile of Arbequina var. EVOO volatile compounds at baseline were characteristic of fresh EVOO, in which the major compounds were aldehydes, alcohols, and esters, the volatile compounds produced by lipoxygenase pathway [8]. (*E*)-2-hexenal, hexanal, 1-hexenol, and (*E*)-2-hexenol were among the most abundant compounds. Following storage of EVOO under different conditions, an increase in volatile compounds such as hexanal and (*E*)-2-hexenal was observed. In addition, there were significant increases in compounds produced by the autoxidation of EVOO, such as propionic and acetic acids and (*E*)-2-nonenal, temperature appearing to be an accelerating factor in the formation of these compounds.

The largest increase in propionic acid from 0.21 to 0.56 mg/kg was observed during the highest storage temperature (C5). The presence of acids in olive oil has been attributed to the oxidation of aldehyde compounds previously formed by the autoxidation of unsaturated fatty acids [41]. Thus, the increase of propionic acid may be the result of autoxidation of linolenic acid through the formation of 16-hydroperoxide, to yield propanal, which by oxidation would then form propionic acid [8,42]. An odor threshold of 0.72 mg/kg was previously reported for propionic acid and imparts defective sensory notes to VOO, such as sour and moldy, although these levels were not reached in our study [7]. 

(*E*)-2-nonenal increased in nearly all storage conditions, with temperature being the most relevant factor for its formation, increasing in the C5 condition from 0.12 mg/kg to 4.58 mg/kg after 12 months of storage. Light was another contributing factor in the formation of (*E*)-2-nonenal, as room temperature with light (C2) reached 1.93 mg/kg and room temperature without light (C3) reached just 0.47 mg/kg after 12 months of storage. Room temperature storage with light was also similar to storage at 30 °C in darkness (C4), with levels reaching 2.03 mg/kg in the latter. Appreciable levels of volatiles were produced in soybean oil containing chlorophyll only under light compared to in the dark [34].

In general, the increased levels of volatile compounds during the storage could be explained by the decomposition of hydroperoxides formed during the autoxidation and photooxidation of olive oils subjected to temperature and light conditions, respectively. Aidos et al. [43] reported that the hydroperoxide decomposition rate of crude herring oil stored at 50 °C in the dark was higher than the formation rate of hydroperoxide. 

The (*E*)-2-nonenal in darkness conditions could be formed from the decomposition of the 9-hydroperoxylinoleic acid via a homolytic cleavage mechanism, resulting in the formation of a hydroxyl radical (HO ^●^) and an alkoxy radical (RO ^●^). β-scission of the C9-C10 linkage would then form an olefin, which would then form the compound (*Z*)-3-nonenal [8], followed by isomerization to yield (*E*)-2-nonenal [44]. In light condition (C2) (*E*)-2-nonenal could be formed from the decomposition of 10-hydroperoxylinoleic acid [30].

The homolytic β-scission was identified as the most important free radical reaction, leading to breakdown products causing flavor deterioration in fats [8,34]. Haze et al. [45] reported the formation of (*E*)-2-nonenal from an omega-7 monounsaturated fatty acid such as palmitoleic acid (16:1ῳ-7), through the formation of 8-hydroperoxypalmitoleic acid. This volatile was reported to have an odor threshold of 0.90 mg/kg [8] and results in defective sensory notes in olive oil such as paper-like, fatty, penetrating, waxy, beany, and rancid notes [8,31]. In our study, the level of (*E*)-2-nonenal did indeed exceed the odor threshold.

Hexanal increased progressively in all storage conditions by between 4 and 44%, reaching its highest levels after 5 months of storage during condition C5, reaching a value of 9.17 mg/kg. This volatile compound has two formation pathways, one through the lipoxygenase pathway during oil processing and the other through an autoxidation mechanism [31]. Both pathways result in the formation of 13-hydroperoxylinoleic acid, which by enzymatic reaction (lipoxygenase pathway) or by decomposition of the hydroperoxide through homolytic cleavage reactions and β-scission of the C12-C13 linkage, would form hexanal. Therefore, the hexanal content at baseline (Fresh EVOO) originates from the milling of the olive and the malaxation process of the olive paste [31], while the increase in hexanal during storage is due to lipid autoxidation [8]. Hexanal contributes to the perception of a sweet-green sensory note in EVOO when its concentration surpasses 0.075 mg/kg [46], but it also contributes to the rancid perception when there is an even higher concentration [47]. Coutelieris and Kanavouras [48] used the concentration of hexanal as a basic indicator of the quality of olive oil stored in various packaging materials and storage conditions. Malheiro et al. [49] reported a decrease in EVOO volatile compounds related to flavor of close to 90% after 12 months of storage at room temperature in Verdeal Transmontana olive oils.

(*E*)-2-hexenal was the most abundant volatile compound in fresh EVOO with a concentration of 15.3 mg/kg of oil and is something lower to previous reports from Youssef et al. [50]. (*E*)-2-hexenal increased gradually in all storage conditions until month 7, with a more accelerated increase until the end of the 12-month period, reaching a value of 19.36 mg/kg when stored at room temperature in darkness (C3). (*E*)-2-hexenal is formed by the lipoxygenase pathway from 13-hydroperoxylinolenic acid and is characterized by a bitter-almond sensory note with an odor threshold of 0.42 mg/kg [41]. During storage, this compound could also be formed from the same precursor by autoxidation. Lobo-Prieto et al. [21] observed a reduction of (*E*)-2-hexenal in VOOs exposed to light.

Ethanol was also present at high concentrations in the oil (14.7 mg/kg), although this was lower than the odor threshold of 30 mg/kg [41]. Ethanol levels decreased throughout the storage period, being particularly significant in the condition of higher temperature (C5), reaching a value of 10.46 mg/kg. High ethanol and free acidity levels in olive oil has been associated with the formation of ethyl esters, which could explain the decline in ethanol over time [51]. In the case of VOO, enzymatic oxidation via lipoxygenase is the main pathway for desirable green and fruity flavors [52], while chemical oxidation is the main contributor to the spoilage of VOO [46].

### 3.6. General Multivariate Characterization of Samples using PCA

Data were organized in a matrix and evaluated by principal component analysis (PCA) (with the data centered) to better elucidate the relationships between the composition of the samples and their storage conditions. The first three components explained 96% of the variability. Figure 3 shows the graph of the samples and variables together (biplot) for component 1 and 3. 

As shown in Figure 3, there are 11 main variables (oxidized phenolic compounds, non-oxidized phenolic compounds, total phenolic compounds, elenoic acid, tocopherols, DAODO (*3,4-DHPEA-EDA*, oxidized form), DALDO (*p*-*HPEA-EDA*, oxidized form), AOAHO *(3,4-DHPEA-EA-AH*, oxidized form), DAOD (*3,4-DHPEA-EDA*), DALD (*p-HPEA-EDA*), and AOAH (*3,4-DHPEA-EA-AH*) that allow for characterization of the samples according to their storage conditions. The samples from storage conditions C1 (−23 °C), C2, and C3 (room temperature) were differentiated from storage conditions C4 and C5 (30 and 40 °C, respectively) through component 3; where the latter were associated with a higher quantity of oxidized phenolic compounds, elenoic acid, DAODO, DALDO, and AOAHO. This result further highlights the increased formation of these compounds under higher storage temperatures. Likewise, the higher content of these components in the oil is accentuated with storage time, as demonstrated for samples from the C4 and C5 conditions when stored for more than 10 and 3 months, respectively.

Conversely, samples from C1, C2, and C3 conditions are associated with higher levels of non-oxidized phenolic compounds, DAOD, AOAH, and DALD, and suggests that lower storage temperatures favor stability and prevent oxidation of the oil components. Furthermore, with the exception of one month of storage, the C2 samples (room temperature with light) were grouped in the lower left quadrant directly associated with non-oxidized phenolic compounds, particularly those with shorter storage time (2 to 5 months). However, these samples are also inversely related to the content of α-tocopherol (upper right quadrant) and highlights the antioxidant action of α-tocopherol in photooxidation, with a dramatic drop throughout the storage period as discussed above. Importantly, the inverse relationship between oxidized phenols and the content of secoiridoids (DALD and DAOD) and α-tocopherols is due to the production of oxidized phenols during oil degradation, which in turn causes a decrease in secoiridoids and tocopherols.

### 3.7. Relationships between Oils Variables and (E)-2-nonenal

Since the PCA not only evidenced the effect of storage conditions on the composition of the oil, but also the effect of storage time on it; (E)-2-Nonenal was selected as a marker of oil quality degradation; as oil deterioration is associated with the appearance of this volatile compound over time. Thus, in order to establish general relationships between the different components of the oil and (E)-2-nonenal, a partial least squares (PLS) regression model was built to visualize how this compound is a function of the independent variables. According to the cross-validation, 4 components were selected, which explained 77% of the variability in Y. Figure 4 shows the graph of the samples and variables jointly (biplot) for component 1 and 2.

As shown in Figure 4, the same 11 variables previously described in PCA stand out in relation to the levels of (E)-2-nonenal in oils. The samples were distributed along the first component in a pattern that considers milder storage conditions and shorter storage times (left half) towards more aggressive conditions and longer storage times (right half). The first group was associated with the presence of α-tocopherol, non-oxidized phenolic compounds, DAOD, AOAH, and DALD. The second group was associated with oxidized phenolic compounds, elenolic acid, DAODO, DALDO, and AOAHO. This relationship is corroborated when analyzing the regression coefficients of the PLS model (Figure 5) for the first component, where the first group of compounds correlates inversely with the content of (E)-2-nonenal and the latter group correlates directly. This group includes the total volatile compounds associated with (E)-2-nonenal. As previously mentioned, the increase in volatile compounds during storage could be explained by the decomposition of hydroperoxides formed during the autoxidation of olive oil subjected to temperature and light conditions. Of particular relevance are the oxidized phenolic compounds that increase in tandem with the volatiles, such as (E)-2-nonenal. This correlation is likely due to the formation of the oxidation product 9- or 10-hydroperoxylinoleic acid, which by decomposition would generate (E)-2-nonenal, as noted earlier.

### 3.8. (E)-2-nonenal as an Oxidation Marker

It is possible that monitoring the degradation progress of EVOO over time could be broadly evaluated measuring (E)-2-nonenal levels. Further, (E)-2-nonenal was found at trace levels in EVOO, and the lipoxygenase cascade—the main biochemical pathway—promotes the formation of C6 volatile compounds from linoleic and linolenic hydroperoxides, rather than the C9 compounds [47].

Thus, (E)-2-nonenal was chosen as a predictor of oxidation. For this purpose, a polynomial fit to the (E)-2-nonenal dataset as a function of time was performed for each storage condition. That is, a linear regression fitted to the dataset was performed, finding significant correlations for conditions C4 (*p* < 0.03) and C5 (*p* < 0.00), obtaining the following equations:C4 Condition, (*E*)-2-nonenal (mg/kg) = 0.128 (month) + 0.5105 (R^2^ = 0.8005) (1)
C5 Condition, (*E*)-2-nonenal (mg/ kg) = 0.797 (month) − 0.6482 (R^2^ = 0.9861) (2)

Previously, it was determined that both hexanal and nonenal are good oxidation indicators for sunflower and palm oils, yielding a good linear relationship with TOTOX [42]. Furthermore, other studies identified the nonanal compound as an early oxidative marker for extra virgin olive oil [47].

Considering that EVOO has an odor threshold for (E)-2-nonenal of 0.9 mg/kg, an oil stored at 30 °C in the dark would reach this odor threshold after 3 months of storage, and if stored at 40 °C in the dark the odor threshold would be reached after just 1.94 months (58 days) of storage. Therefore, under the above conditions a consumer could perceive a fatty or rancid aroma in the oil, indicating an off-flavor, and the oil would lose its extra virgin category [2].

## 4. Conclusions

Principal component analysis allowed to better understand the relationships between the composition of the olive oils and the storage conditions. Eleven variables were identified (oxidized phenolic compounds, non-oxidized phenolic compounds, total phenolic compounds, elenolic acid, α-tocopherol, *3,4-DHPEA-EDA* oxidized, *p*-*HPEA-EDA* oxidized, *3,4-DHPEA-EA-AH* oxidized, *3,4-DHPEA-EDA*, *p*-*HPEA-EDA*, and *3,4-DHPEA-EA-AH*) that allowed for characterization of the samples according to their storage conditions. The EVOOs stored at lower temperatures under conditions C1, C2, and C3 differed from the oils stored at higher temperatures (conditions C4 and C5). The higher temperature conditions were associated with increased levels over time of compounds derived from the hydrolysis and oxidation of secoiridoid derivatives. EVOOs stored under lower temperature conditions (C1, C2, and C3) were associated with a higher quantity of secoiridoid derivatives and non-oxidized phenolic compounds, indicating that lower storage temperatures limited the oxidation of the oil components. The oils stored in condition C2 were inversely related to the content of α-tocopherols, highlighting the antioxidant action of α-tocopherol in response to photooxidation.

Utilizing a regression on partial least squares, a model was built that described the formation of (*E*)-2-nonenal as a function of the 11 independent variables described in the PCA, indicating (*E*)-2-nonenal as a marker of EVOO quality over time. Linear regressions of (*E*)-2-nonenal were determined as a function of time, with significance for conditions C4 and C5. Therefore, early oxidation in EVOO could be monitored by measuring (*E*)-2-nonenal in the oil. 

Through an analytical focus, this study provides scientific evidence that storage conditions are highly relevant to ensure the quality of EVOO. As such, storage conditions are an important factor to consider when exporting oils to destination markets that have higher average room temperatures than those recommended for EVOO storage. Lack of consideration for storage conditions could result in the appearance of sensory defects in the oil and thus in a loss of EVOO category.

## Figures and Tables

**Figure 1 foods-10-02161-f001:**
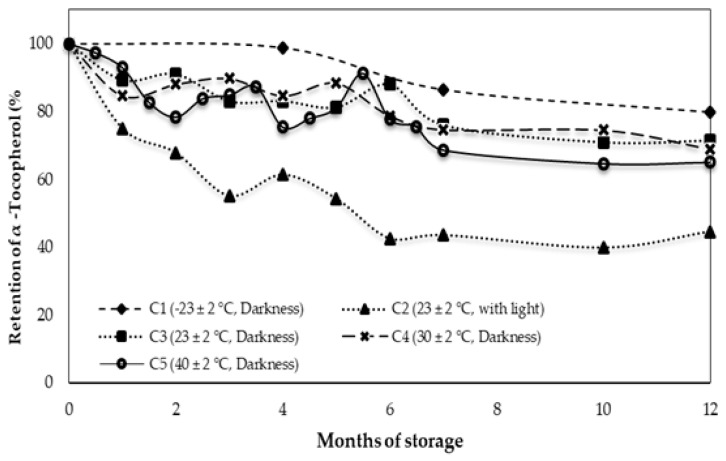
Evolution of α- tocopherol, during 12 months of storage for different storage conditions.

**Figure 2 foods-10-02161-f002:**
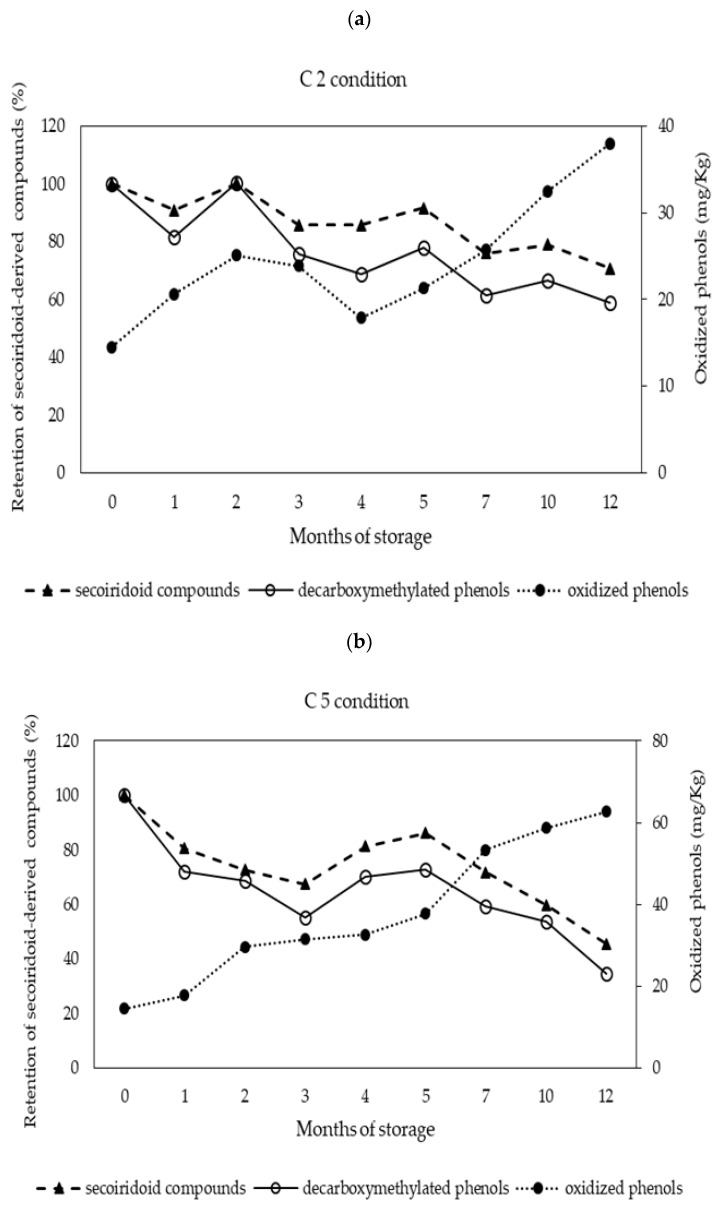
Evolution of secoiridoid compounds, decarboxymethylated phenols, and oxidized phenols during 12 months of storage for the conditions (**a**) C2 (23 ± 2 °C, with light) and (**b**) C5 (40 ± 2 °C, in darkness).

**Figure 3 foods-10-02161-f003:**
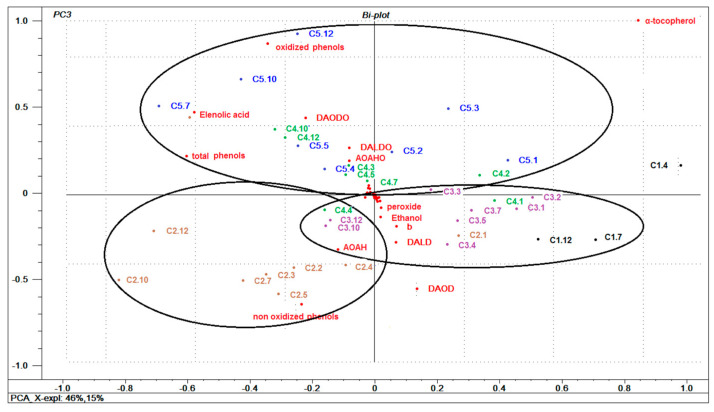
Multivariate general characterization of oil samples by PCA. Plots of samples and variables. Sample are labeled by storage conditions (C1: stored at −23 ± 2 °C in darkness; C2: 23 ± 2 °C, with light; C3: 23 ± 2 °C in darkness; C4: 30 ± 2 °C in darkness; C5: 40 ± 2 °C in darkness); and storage time (from 1 to 12 months). Abbreviations: AOAH: *3,4-DHPEA-EA-AH*; AOAHO: *3,4-DHPEA-EA-AH* oxidized; DAOD: *3,4-DHPEA-EDA*; DAODO: *3,4-DHPEA-EDA*, oxidized; DALD: *p*-*HPEA-EDA*; DALDO: *p*-*HPEA-EDA*, oxidized.

**Figure 4 foods-10-02161-f004:**
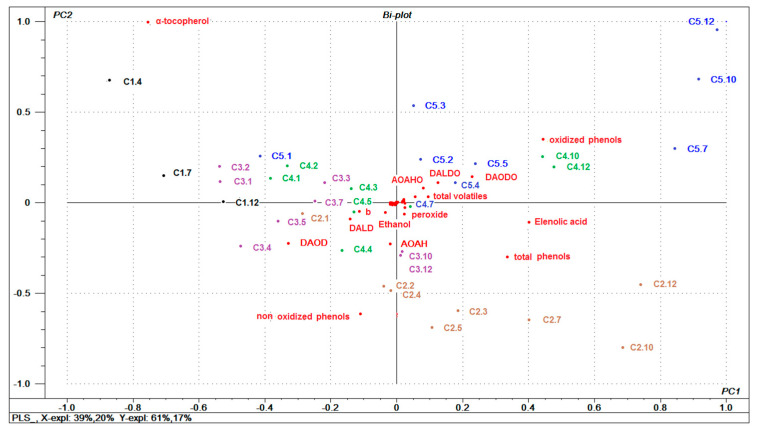
Relationships between oils variables and (*E*)-2-nonenal by PLSA. Plots of samples and variables. Sample are labeled by storage conditions. Abbreviations see Figure 3.

**Figure 5 foods-10-02161-f005:**
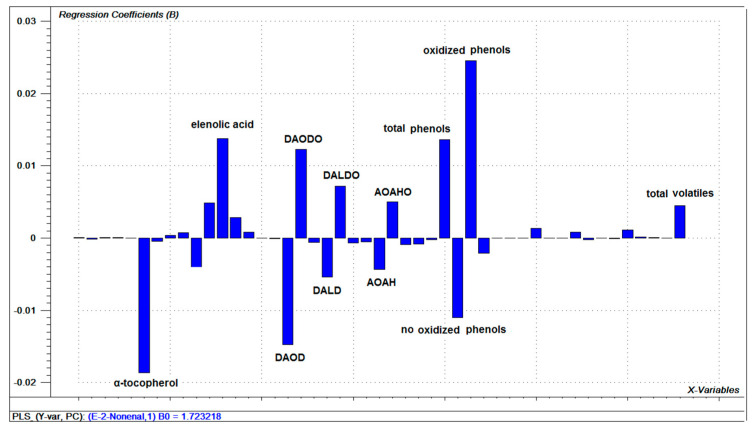
Regression vector of the PLS model for relation between oils variables and (*E*)-2-nonenal.

**Table 1 foods-10-02161-t001:** Variation of free fatty acid, peroxide value, K232 and K270 in Arbequina variety EVOO, subjected to different temperature and light condition during 12 months of storage.

	Free Fatty Acid (% m/m Oleic Acid)							K232 (Absorbance Units)					
Month	C1	C2	C3	C4	C5	*p-*Value	Month	C1	C2	C3	C4	C5	*p-*Value
0	0.17 ± 0.00	0.17 ± 0.00	0.17 ± 0.00	0.17 ± 0.00	0.17 ± 0.00	-	0	1.55 ± 0.01	1.55 ± 0.01	1.55 ± 0.01	1.55 ± 0.01	1.55 ± 0.01	-
2	-	0.19 ± 0.01 ab	0.18 ± 0.00 a	0.19 ± 0.00 ab	0.20 ± 0.00 b	*	2	-	1.69 ± 0.08 ab	1.66 ± 0.02 a	1.84 ± 0.02 bc	1.88 ± 0.09 c	*
3	-	0.19 ± 0.00 a	-	0.21 ± 0.00 b	0.22 ± 0.01 b	*	3	-	1.72 ± 0.04 a	1.85 ± 0.07 ab	1.90 ± 0.07 ab	1.94 ± 0.10 b	*
4	0.17 ± 0.01 a	0.16 ± 0.01 a	0.17 ± 0.01 a	0.18 ± 0.01 a	0.21 ± 0.01 b	**	4	1.76 ± 0.08 a	1.81 ± 0.11 ab	1.99 ± 0.03 b	1.95 ± 0.02 ab	1.88 ± 0.09 ab	*
5	-	0.19 ± 0.01 a	0.18 ± 0.01 a	0.19 ± 0.01 a	0.20 ± 0.00 b	*	5	-	1.83 ± 0.07 a	1.93 ± 0.07 a	1.92 ± 0.01 a		+
6	-	0.17 ± 0.00 a	0.17 ± 0.01 a	0.18 ± 0.01 b	0.21 ± 0.01 c	**	6	-	1.84 ± 0.00 a	2.05 ± 0.01 a	1.95 ± 0.23 a		+
7	0.18 ± 0.01 a	0.19 ± 0.01 ab	0.20 ± 0.01 ab	0.21 ± 0.00 b	0.28 ± 0.02 c	**	7	1.68 ± 0.11 a	1.78 ± 0.04 a	1.89 ± 0.09 a	1.85 ± 0.06 a		+
10	-	0.19 ± 0.01 a	0.18 ± 0.01 a	0.22 ± 0.01 b	0.29 ± 0.01 c	**	10	-	1.81 ± 0.02 a	1.79 ± 0.10 a	1.86 ± 0.05 a	1.98 ± 0.01 b	*
12	0.18 ± 0.01 a	0.19 ± 0.00 b	0.19 ± 0.01 b	0.23 ± 0.00 c	0.32 ± 0.00 d	**	12	1.60 ± 0.09 a	1.80 ± 0.00 ab	2.03 ± 0.05 b	1.84 ± 0.06 a	1.96 ± 0.03 ab	*
**Peroxide Value (meq O_2_/kg)**							**K270 (Absorbance Units)**						
**Month**	**C1**	**C2**	**C3**	**C4**	**C5**	***p-*Value**	**Month**	**C1**	**C2**	**C3**	**C4**	**C5**	***p-*Value**
0	2.9 ± 0.1	2.9 ± 0.1	2.9 ± 0.1	2.9 ± 0.1	2.9 ± 0.1	-	0	0.09 ± 0.01	0.09 ± 0.01	0.09 ± 0.01	0.09 ± 0.01	0.09 ± 0.01	-
2	-	5.5 ± 0.2 c	4.4 ± 0.0 a	5.1 ± 0.2 bc	4.8 ± 0.1 ab	**	2		0.11 ± 0.01 c	0.08 ± 0.01 a	0.09 ± 0.00 ab	0.11 ± 0.01 bc	**
3	-	6.7 ± 0.5 b	4.6 ± 0.1 a	4.9 ± 0.0 a	5.3 ± 0.6 a	**	3		0.12 ± 0.00 c	0.09 ± 0.00 a	0.10 ± 0.01 ab	0.11 ± 0.01 b	**
4	3.5 ± 0.1 a	6.7 ± 0.1 d	4.8 ± 0.1 b	5.2 ± 0.2 c	4.6 ± 0.0 b	**	4	0.09 ± 0.01 a	0.13 ± 0.00 c	0.09 ± 0.01 a	0.10 ± 0.00 a	0.12 ± 0.00 b	**
5	-	7..2 ± 0.7 b	5.3 ± 0.2 a	5.1 ± 0.1 a	5.0 ± 0.2 a	**	5		0.13 ± 0.01 b	0.09 ± 0.00 a	0.10 ± 0.00 a	0.12 ± 0.01 b	**
6	-	7.4 ± 0.00 b	5.3 ± 0.4 a	5.1 ± 0.0 a	4.4 ± 0.0 a	*	6		0.13 ± 0.00 c	0.09 ± 0.01 a	0.11 ± 0.01 b	0.13 ± 0.00 ^c^	**
7	3.4 ± 0.1 a	7.9 ± 0.01 c	5.6 ± 0.8 b	5.1 ± 0.3 b	4.3 ± 0.6 ab	**	7	0.09 ± 0.01 a	0.13 ± 0.01 c	0.09 ± 0.00 a	0.11 ± 0.01 b	0.15 ± 0.01 ^c^	**
10	-	8.5 ± 0.8 c	5.1 ± 0.1 ab	5.7 ± 0.2 b	4.0 ± 0.4 a	**	10		0.15 ± 0.01 c	0.09 ± 0.01 a	0.12 ± 0.00 b	0.17 ± 0.01 ^d^	**
12	3.3 ± 0.1 a	8.2 ± 0.8 d	5.9 ± 0.2 c	5.3 ± 0.1 bc	4.4 ± 0.2 b	**	12	0.09 ± 0.00 a	0.15 ± 0.01 c	0.10 ± 0.01 a	0.13 ± 0.00 b	0.18 ± 0.01 ^d^	**

Abbreviation: C1, −23 ± 2 °C in darkness; C2, 23 ± 2 °C with light; C3, 23 ± 2 °C in darkness; C4, 30 ± 2 °C in darkness; C5 40 ± 2 °C in darkness. Different letters in the same row indicate significant differences between storage conditions. *p*-Value (HSD Tukey) *: *p* < 0.05; **: *p* < 0.001; +: no significant difference.

**Table 2 foods-10-02161-t002:** Content of phenolic compounds, α-tocopherol, and antioxidant capacity after 12 months of storage under different conditions.

Phenolic Compound (mg/kg)	Fresh EVOO (Month 0)	C1	C2	C3	C4	C5
Elenolic acid	76.5 ± 1.8 a	73.1 ± 3.6 a	107.9 ± 0.4 c	107.4 ± 4.9 c	113.6 ± 5.4 c	98.5 ± 3.5 b
Hydroxytyrosol	3.1 ± 3.0 a	3.2 ± 0.1 a	6.0 ± 0.3 c	5.1 ± 0.1 b	6.9 ± 0.1 d	8.8 ± 0.3 e
Tyrosol	5.1 ± 0.3 a	5.0 ± 0.2 a	6.0 ± 0.3 c	5.4 ± 0.2 ab	5.7 ± 0.1 bc	6.5 ± 0.1 d
Vanillic acid	0.4 ± 0.0 b	0.3 ± 0.0 a	0.3 ± 0.0 ab	0.3 ± 0.0 a	0.3 ± 0.0 a	0.3 ± 0.0 a
*p*-Coumaric acid	1.8 ± 0.1 c	0.9 ± 0.1 b	0.7 ± 0.0 a	0.8 ± 0.0 ab	0.8 ± 0.0 a	0.8 ± 0.0 ab
*3,4-DHPEA-EDA*	74.7 ± 3.3 e	64.9 ± 1.6 d	42.7 ± 0.9 b	52.2 ± 0.7 c	40.4 ± 0.8 b	25.3 ± 1.9 a
*3,4-DHPEA-EDA,* oxidized form	2.7 ± 0.3 a	2.4 ± 0.2 a	15.8 ± 1.3 d	6.8 ± 0.4 b	14.1 ± 0.3 c	25.3 ± 0.7 e
*3,4-DHPEA-EDA-DOA*	3.5 ± 1.0 b	1.5 ± 0.2 a	0.6 ± 0.1 a	0.7 ± 0.0 a	0.5 ± 0.1 a	0.4 ± 0.1 a
*p-HPEA-EDA*	34.7 ± 2.7 e	29.1 ± 0.7 d	21.8 ± 0.9 b	24.9 ± 1.0 c	21.0 ± 0.5 b	12.7 ± 0.5 a
*p-HPEA-EDA,* oxidized form	2.3 ± 0.2 a	1.8 ± 0.0 a	8.3 ± 0.9 c	4.6 ± 0.3 b	9.4 ± 0.3 d	17.2 ± 0.3 e
Pinoresinol ^a^	15.8 ± 1.2 c	15.1 ± 0.8 bc	14.2 ± 0.5 abc	13.7 ± 0.3 ab	12.8 ± 0.2 a	12.7 ± 1.1 a
*p-HPEA-EDA-DLA*	4.3 ± 0.4 ab	4.6 ± 0.3 b	4.0 ± 0.3 ab	3.7 ± 0.1 ab	3.5 ± 0.3 a	4.2 ± 0.8 ab
*3,4-DHPEA-EA-AH,* oxidized form	9.6 ± 0.7 a	10.8 ± 0.4 ab	13.9 ± 0.8 c	11.0 ± 0.6 b	15.5 ± 0.5 d	20.3 ± 0.9 e
*3,4-DHPEA-EA-AH*	27.8 ± 2.9 b	30.2 ± 0.3 bc	33.3 ± 0.7 c	38.7 ± 1.4 d	30.3 ± 1.2 bc	23.5 ± 0.6 a
Luteolin	7.7 ±0.7 d	8.3 ± 0.2 d	6.1 ± 0.2 bc	6.6 ± 0.2 c	5.4 ± 0.1 ab	5.2 ± 0.2 a
Apigenin	2.4 ± 0.3 a	2.8 ± 0.2 b	2.4 ± 0.1 a	2.4 ± 0.1 ab	2.3 ± 0.0 a	2.3 ± 0.1 a
Methyl luteolin	1.3 ± 0.2 a	1.5 ± 0.1 b	1.3 ± 0.1 a	1.3 ± 0.0 ab	1.2 ± 0.0 a	1.2 ± 0.0 a
Total phenols	273.5 ± 8.2 bc	255.7 ± 5.1 a	285.3 ± 4.1 c	285.6 ± 6.2 c	283.5 ± 7.8 c	265.2 ± 7.1 ab
non-oxidized phenols	259.0 ± 7.5 cd	240.6 ± 4.9 b	247.3 ± 1.8 bc	263.3 ± 5.8 d	244.5 ± 7.7 b	202.5 ± 5. 6 a
Oxidized phenols	14.5 ± 0.9 a	15.0 ± 0.5 a	38.0 ± 2.8 c	22.3 ± 0.9 b	39.0 ± 0.3 c	62.8 ± 1.8 d
α-Tocopherol (mg*/*kg)	180.2 ± 2.4 d	144.0 ± 10.6 c	80.3 ± 4.0 a	128.8 ± 2.6 bc	124.3 ± 0.6 b	117.3 ± 1.2 b
ORAC (μmol TE*/*g)	5.5 ± 0.2 b	3.7 ± 0.3 a	3.6 ± 0.2 a	4.1 ± 0.4 a	3.9 ± 0.3 a	3.9 ± 0.4 a

Note: Different letters in the same row indicate significant differences between storage conditions (*p* < 0.05; HSD Tukey). Values are mean ± SD (*n* = 3). ^a^: Mixed with 1-acetoxy-pinoresinol. Abbreviations: *3,4-DHPEA-EDA*, dialdehydic form of decarboxymethyl oleuropein aglycon. *3,4-DHPEA-EDA-DOA*, dialdehydic form of oleouropein aglycon. *p-HPEA-EDA*, dialdehydic form of decarboxymethyl ligstroside aglycon. *p-HPEA-EDA-DLA*, dialdehydic form of ligstroside aglycon. *3,4-DHPEA-EA-AH*, aldehyde, and hydroxylic form of oleuropein aglycone.

**Table 3 foods-10-02161-t003:** Content of volatile compounds after 12 months of storage under different conditions.

Volatile Compounds	Fresh EVOO (Month 0)	C1	C2	C3	C4	C5	Sensory Atributes	Odor Thresholdin Oil (mg/kg)
Ethanol	14.74 ± 0.71 bcd	15.81 ± 1.57 d	14.37 ± 0.68 bc	15.68 ± 0.37 cd	13.2 ± 0.51 b	10.46 ± 0.27 a	Apple, sweet	30
Ethylpropanoate	0.1 ± 0.01 b	0.1 ± 0.01 b	0.09 ± 0.01 ab	0.1 ± 0.01 b	0.09 ± 0.01 ab	0.08 ± 0.01 a	Strawberry, apple, fruity	0.10
4-methyl-pentan-2-one	0.13 ± 0.01 a	0.16 ± 0.01 b	0.16 ± 0.01 b	0.16 ± 0.01 b	0.16 ± 0.01 b	0.16 ± 0.01 b	Strawberry, fruity, sweet, ethereal	0.30
Butylacetate	0.05 ± 0.01 b	0.04 ± 0.00 ab	0.04 ± 0.00 ab	0.04 ± 0.01 ab	0.05 ± 0.01 b	0.04 ± 0.01 a	Green, fruity, pungent, sweet	0.10
Hexanal	6.26 ± 0.1 a	6.52 ± 0.01 a	6.97 ± 0.34 b	7.08 ± 0.11 b	7.66 ± 0.04 c	8.6 ± 0.16 d	Green apple, grass	0.075
2-methyl-butan-1-ol	0.12 ± 0 b	0.12 ± 0.01 b	0.11 ± 0.01 b	0.12 ± 0.01 b	0.11 ± 0.01 b	0.09 ± 0 a	Winey, spicy	0.48
3-methyl-butan-1-ol	0.27 ± 0 bc	0.29 ± 0.01 d	0.27 ± 0.01 c	0.28 ± 0.01 cd	0.26 ± 0.01 b	0.21 ± 0.01 a	Woody, sweet	0.10
(*E*)-2-hexenal	15.28 ± 0.24 a	18.7 ± 0.25 b	18.34 ± 1.2 b	19.36 ± 0.53 b	18.94 ± 0.47 b	18.26 ± 0.39 b	Bitter almonds, green- fruity	0.42
Hexan-1-ol	5.76 ± 0.18 bc	5.85 ± 0.04 bc	5.89 ± 0.4 c	5.55 ± 0.14 bc	5.5 ± 0.13 b	5.0 ± 0.15 a	Fruity, soft, aromatic	0.40
(*E*)-2-nonenal	0.12 ± 0.12 a	0.21 ± 0.15 a	1.93 ± 0.29 b	0.48 ± 0.29 a	2.03 ± 0.56 b	4.58 ± 1.07 c	Fatty, rancid, paper-like, penetrating, waxy, beany	0.90
(*E*)-2-hexen-1-ol	7.51 ± 0.05 a	7.85 ± 0.03 b	7.76 ± 0.10 b	7.88 ± 0.09 b	7.79 ± 0.1 b	7.49 ± 0.07 a	Green grass, leaves, fruity, astringent, bitter	5.00
Acetic acid	6.15 ± 0.27 a	6.58 ± 0.22 a	7.24 ± 0.33 b	7.27 ± 0.19 b	7.56 ± 0.14 b	8.04 ± 0.24 c	Sour, vinegary	0.50
Propionic acid	0.21 ± 0.02 a	0.2 ± 0.01 a	0.26 ± 0.02 b	0.38 ± 0.03 c	0.47 ± 0.02 d	0.56 ± 0.01 e	Pungent, sour, mould	0.72
Octan-1-ol	0.18 ± 0.01 a	0.14 ± 0.01 a	0.15 ± 0.04 a	0.13 ± 0.04 a	0.17 ± 0.04 a	0.18 ± 0.03 a	-	-
Butanoic acid	0.33 ± 0.01 c	0.17 ± 0.02 ab	0.16 ± 0.01 ab	0.15 ± 0.02 a	0.18 ± 0.02 ab	0.2 ± 0.03 b	Rancid, cheese, sweat	0.65
Total Volatiles	57.23 ± 0.60 a	62.74 ± 1.99 b	63.74 ± 1.07 b	64.64 ± 0.93 b	64.17 ± 1.26 b	63.94 ± 1.66 b		

Note: Values are mean ± SD (*n* = 3). Different letters in the same row indicate significant differences between storage conditions (*p* < 0.05; HSD Tukey).

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
