# Peer review of "Effect of Storage Conditions on the Quality of Arbequina Extra Virgin Olive Oil and the Impact on the Composition of Flavor-Related Compounds (Phenols and Volatiles)"

_foods, 2021, doi:10.3390/foods10092161_

Round 1
Reviewer 1 Report
The article is in general interesting and well written, however some suggestions are reported below:
In my opinion authors should better highlight and discuss the differences between their applied storage conditions (specifically storage conditions at -23°C, 30°C and 40°C, so close to possible temperature reached during trasportation of oils during winter/summer seasons) respect to other cited studies in literature, in particular the references 23-26 in which changes in volatile and phenolic profiles were also considered.
The reasons for the selection of the five studied storage conditions should be explained and discussed taking into consideration the works published in literature.
2.4. Determination of Phenolic Compounds
Elenolic acid has not absorption at 280 nm or 335 nm but in Table 2 its concentration is reported so, please, add more information on its determination.
Nomenclature as "(E)-2-Hexenal" instead of "E-2-hexenal" it would be better in all text and Table 3 for all volatile compounds.
Table 3: Please, change "," with "." in Total Volatiles data
Author Response
Response to Reviewer 1 Comments
Thank you very much for reviewing our manuscript. We have made the suggested modifications, considering your comments.
The article is in general interesting and well written, however some suggestions are reported below:
Point 1: In my opinion authors should better highlight and discuss the differences between their applied storage conditions (specifically storage conditions at -23°C, 30°C and 40°C, so close to possible temperature reached during trasportation of oils during winter/summer seasons) respect to other cited studies in literature, in particular the references 23-26 in which changes in volatile and phenolic profiles were also considered.
Answer. We appreciate this comment. For a better understanding of the storage conditions applied in other research, a sentence explaining the conditions used in the publications 23 -26 was added in the introduction.
Point 2: The reasons for the selection of the five studied storage conditions should be explained and discussed taking into consideration the works published in literature.
Answer: In section 2.2.1. Storage Conditions a paragraph was added explaining why the five storage conditions were selected.
Point 3: 2.4. Determination of Phenolic Compounds Elenolic acid has not absorption at 280 nm or 335 nm but in Table 2 its concentration is reported so, please, add more information on its determination.
Answer: The explanation of the determination of elenolic acid was added.
Point 4: Nomenclature as "(E)-2-Hexenal" instead of "E-2-hexenal" it would be better in all text and Table 3 for all volatile compounds.
Answer: E-2-hexanal was changed by (E)-2 hexenal following the format of other Journal Foods publications.
Point 5: Table 3: Please, change "," with "." in Total Volatiles data
Answer: The change was done.
Reviewer 2 Report
Manuscript # FOODS1339032
“Effect of storage conditions on the quality of Arbequina extra virgin olive oil and the impact on the composition of flavor-related compounds (Phenols and volatiles)."
By Leeanny Caipo et al.
The paper presents an evaluation on the flavor of Arbequina extra virgin olive oil (EVOO)on different quality parameters of the EVOOs. The manuscript is well written. However, in my opinion the conclusion reported in which the storage conditions can affect the quality of the product is obvious. In addition, a drawback of the work is the lacking of sensory analysis that is a quality parameter also included in EU legal standard to define EVOO and that could reinforce the results about the definition of 2-nonenal as marker of early stage of oil quality degradation. Details: the name of the plant species should be in Latin binomial, please homogenise along the text the name of E-2-nonenal and tran-2-nonenal. E-nonenal the E should be in italics, as well as those of other components i.e. hexenal.
Author Response
Response to Reviewer 2 Comments
Thank you very much for reviewing our manuscript. We have made the suggested modifications, considering your comments.
The paper presents an evaluation on the flavor of Arbequina extra virgin olive oil (EVOO)on different quality parameters of the EVOOs. The manuscript is well written.
Point 1: However, in my opinion the conclusion reported in which the storage conditions can affect the quality of the product is obvious.
Answer: The paper highlights the importance of storage conditions, especially in destination markets with higher average temperatures than those recommended for oil storage. An important conclusion is that “Utilizing a regression on partial least squares, a model was built that described the formation of (E)-2-nonenal as a function of the 11 independent variables described in the PCA, indicating (E)-2-nonenal as a marker of EVOO quality over time. Linear regressions of (E)-2-nonenal were determined as a function of time, with significance for conditions C4 and C5. Therefore, early oxidation in EVOO could be monitored by measuring (E)- 2-nonenal in the oil”.
Point 2: In addition, a drawback of the work is the lacking of sensory analysis that is a quality parameter also included in EU legal standard to define EVOO and that could reinforce the results about the definition of 2-nonenal as marker of early stage of oil quality degradation.
Answer. We appreciate this comment, but for this study we are only considering chemical analysis. For the sensory evaluation, a panel of evaluators trained and certified by the IOC is required, in addition to a significant amount of oil of at least 500 cc per sample, unfortunately for logistical and cost reasons we did not have access to a panel. However, for our future studies we will try to include this type of evaluation.
Point 3: Details: the name of the plant species should be in Latin binomial
Answer: The name of the plant was removed on the advice of another reviewer.
Point 4: please homogenise along the text the name of E-2-nonenal and tran-2-nonenal. E-nonenal the E should be in italics, as well as those of other components i.e. hexenal.
Answer: E-2-, trans-2- was chaged by (E)-2- and homogenised along the text.
Reviewer 3 Report
Review: Foods Manuscript ID 1339032
The manuscript presents a detailed chemical analysis of olive oils which has been subjected to five different storage conditions. Overall, the experimental question is relevant, but the experimental design and data analysis is highly lacking. Regarding the experimental design, the inclusion of three additional treatments: 30C/Light, 40C/Light, and “Cold”/Light, would have allowed for an evaluation of light and temperature interactions. Quantifying the interactive effect would be beneficial to explain the chemical behavior of; for example, the decline in alpha-tocopherol. Figure 1 shows a substantial decrease at RT/light, but is the decline a result of light or temperature? With a proper design this could likely be answered, but without a full design all one has is speculation. Thus, with the presented data why include a light treatment? A few sentences as to why these 5 treatments were selected would help the reader.
Regarding data analysis – The manuscript does not seem to have a cohesive data analysis plan. Is the intent of the manuscript to model the decline of a given chemistry? Or simply show general trends? Additionally, the application of PLS does not seem to fit the collected data. It seems that you use all the measured chemistry as predictor variables to model one compound as a response variable. Is this correct? How and why PLS is applied needs clarification.
Comments by line:
Line 36 – Olive oil is the product not the plant
Line 260 – “Statistically higher” – by what metric?
Line 346 – 347 From Figure 1, it is difficult to make this statement. There looks to be no difference among C1, C3, C4, and C5, but C2 is likely different.
Line 346 – 352 Consider plotting the first order kinetics. If you model the kinetics, you can show that the multiple curves are statistically different. If you choose not to show the kinetic model fit, the reference to the kinetic order should be removed.
Table 2 – The table caption indicated n=4, is this correct? The methods indicate only indicate triplicate at line 115
Line 405 – 409 how did you calculate the significant difference?
Figure 3 – Shows PC1 vs PC3? Are you intending for this? it is usually standard to plot PC1 vs PC2. Also, unscrambler produces graphics that are difficult to read. Please consider using a secondary plotting program, e.g., Sigma Plot. Consider using color and shape to plot the treatment points.
Line 580 – This sentence is not clear.
See comments above for PLS.
Author Response
Response to Reviewer 3 Comments
Thank you very much for reviewing our manuscript. We have made the suggested modifications, considering your comments.
Point 1: The manuscript presents a detailed chemical analysis of olive oils which has been subjected to five different storage conditions. Overall, the experimental question is relevant, but the experimental design and data analysis is highly lacking. Regarding the experimental design, the inclusion of three additional treatments: 30C/Light, 40C/Light, and “Cold”/Light, would have allowed for an evaluation of light and temperature interactions.
Answer: The evaluation of light and temperature interactions was performed at room temperature with condition 2 (C2): room temperature (23 ± 2 °C) with light and condition 3 (C3): room temperature (23 ± 2 °C) in the dark. A noticeable effect of light on alpha-tocopherol decay was observed.
Carrying out a study with light at the other temperature conditions would have increased the number of samples to be analysed excessively, as well as the time and cost required for the analysis of all the parameters determined in the study.
Point 2: Quantifying the interactive effect would be beneficial to explain the chemical behavior of; for example, the decline in alpha-tocopherol. Figure 1 shows a substantial decrease at RT/light, but is the decline a result of light or temperature? With a proper design this could likely be answered, but without a full design all one has is speculation.
Answer: As explained above, conditions C2 and C3 were performed at room temperature, condition C2 was exposed to light and condition C3 to darkness. Therefore substantial decrease at RT/light for alpha tocopherol shown in Figure 1 is due to the effect of light, as in the C3 condition in the dark alpha tocopherol decreased slightly. Therefore there would be no speculation.
Point 3: Thus, with the presented data why include a light treatment?
Answer: Light treatment was included to compare its effect versus dark storage at room temperature. Also because the oil contains chlorophyll pigments and therefore the oxidation mechanisms involved are different. This is explained in point 2.2.1. Storage Conditions
Point 4: A few sentences as to why these 5 treatments were selected would help the reader.
Answer: In point 2.2.1. Storage Conditions an explanation of the selection of the 5 treatments was added.
Point 5: Regarding data analysis – The manuscript does not seem to have a cohesive data analysis plan. Is the intent of the manuscript to model the decline of a given chemistry? Or simply show general trends? Additionally, the application of PLS does not seem to fit the collected data. It seems that you use all the measured chemistry as predictor variables to model one compound as a response variable. Is this correct? How and why PLS is applied needs clarification.
Answer: Indeed, we wanted to show which components of the oil are related to E-2-nonenal through a PLS general model, being this compound a marker of the quality of the product. It is not convenient to rule out any compound a priori and therefore all those that were determined are included. However, the model clearly indicated that only some of them are related to the chosen quality marker. This was expressed in the sentence lines 545-551.
Comments by line:
Point 6: Line 36 – Olive oil is the product not the plant
Answer: The plant name was deleted.
Point 7: Line 260 – “Statistically higher” – by what metric?
Answer: p value (p <0.05) was added. The statistical methodology carried out for the treatment of the data is explained in the point 2.9. Statistical Analysis. The paragraph is as follows “The results are presented as means ± standard deviation. The data was statistically analyzed using an unpaired Student's t-test and one-way ANOVA to compare the means and a Mann-Whitney test to compare the medians. In all statistical tests p values lower than 0.05 were considered statistically significant. The analyses were performed using the Statgraphic XV software (Rockville, MD, USA).”
Point 8: Line 346 – 347 From Figure 1, it is difficult to make this statement. There looks to be no difference among C1, C3, C4, and C5, but C2 is likely different.
Answer: This sentence refers to the behaviour of the C5 condition (40 ± 2°C, in darkness) with respect to its initial α-tocopherol content (180 mg/kg). During storage at 40 °C the EVOO lost 35% of α-tocopherol, reaching 117 mg/kg, which was significant. The phrase was improved. Line 313-314.
Point 9: Line 346 – 352 Consider plotting the first order kinetics. If you model the kinetics, you can show that the multiple curves are statistically different. If you choose not to show the kinetic model fit, the reference to the kinetic order should be removed.
Answer: the reference to the kinetic order was removed and reference 39 was eliminated.
Point 10: Table 2 – The table caption indicated n=4, is this correct? The methods indicate only indicate triplicate at line 115
Answer: The table caption in Table 2 was corrected n = 3.
Point 11: Line 405 – 409 how did you calculate the significant difference?
Answer: The data was statistically analyzed using an unpaired Student's t-test and one-way ANOVA to compare the means and a Mann-Whitney test to compare the medians.
Point 12: Figure 3 – Shows PC1 vs PC3? Are you intending for this? it is usually standard to plot PC1 vs PC2.
Answer: Although the first two components explain a greater proportion of the variability of the data; it was chosen to plot the components 1 vs 3 since the distribution of the samples was more clearly observed than for the plot of components 1 and 2
Point 13: Also, unscrambler produces graphics that are difficult to read. Please consider using a secondary plotting program, e.g., Sigma Plot. Consider using color and shape to plot the treatment points.
Answer: Figures 3 and 4 were corrected by including the label only of the variables that allowed characterizing the samples. The points representing the non-relevant variables (in the center of the axes) were kept, but without their corresponding labels. In this way a clearer figure is presented. Moreover, different color were used for the treatment points
Point 14: Line 580 – This sentence is not clear.
Answer: The sentence was clarified in line 554-551.
Point 15: See comments above for PLS.
Answer: This was answered in Response 5.
Round 2
Reviewer 3 Report
NA